# Selection of a Sustainable Structural Beam Material for Rural Housing in Latin América by Multicriteria Decision Methods Means

**Juan Francisco Nicolalde** [1,*] **, John Yaselga** [1] **and Javier Martínez-Gómez** [1,2,3]

1    Facultad de Ingeniería y Ciencias Aplicadas, Universidad Internacional SEK, Quito 170302, Ecuador;
     jyaselga.mec@uisek.edu.ec (J.Y.); javier.martinez@uisek.edu.ec (J.M.-G.)
2    Instituto de Investigación Geológico y Energético (IIGE), Quito 170518, Ecuador
3    Departamento de Teoría de la Señal y Comunicación, Universidad de Alcalá, 28801 Madrid, Spain
*    Correspondence: juan.nicolalde@uisek.edu.ec

**Abstract:** Latin America presents the need of more accessible housing in the rural areas of the region; however, problems that this populations have to face are cost, accessibility and seismic events that leave structural damages in the buildings of the population, raising the need of resisting permanent housing or temporary shelter. Moreover, the conventional steel materials are not always affordable, which makes us consider the utilization of alternative sustainable materials that are easier to reach and with a seismic resistance. In this sense, the conditions of an earthquake are simulated, specialized software selects and recreates candidate materials and multicriteria decision methods make the final selection of the best material to be evaluated in a simulation. In this way, it has been found that a beam must resist an effort of 5.96 MPa, candidate materials are composites and naturals, and considering mechanical, environmental and price aspects, the decision methods chose the bamboo as the best material. Lastly, another simulation validates the selection, allowing us to conclude that the multicriteria decision methods used succeeded to find the best alternative.

**Keywords:** material selection; naturals; composites; multicriteria decision; simulation; earthquake

## 1. Introduction

In Latin American countries, it is common to find social housing in rural areas since these are built for the population with lower resources [1], which are made with conventional materials such as concrete, hollow blocks, clay blocks beams and vault slabs [1]. However, the construction sector of Latin America has seen that the use of imported materials has a negative impact on the cost of the houses, something that could be changed with the utilization of native and alternative materials [2]. Nevertheless, prices are not the only problematic that social housing has to face, also, since the Oceanic Nazca plate subducts eastward beneath South America from Chile to Panama, the houses of the regions have to resist some of the strongest earthquakes in the world. For example, in 2016, the Ecuadorian shores suffered a 7.8 earthquake in the Richter Scale, making it the worst in the last years [3], where buildings suffered several infrastructure damages. In this sense, the telluric forces are an important parameter to consider for an alternative material that can be useful in social housing. In this way, taking as reference urban areas, traditionally the support of slabs and roofs are widely done with steel decking since the suppliers have promoted technical–constructive–economic advantages and it showed small damages and good energy dissipation [4]. Furthermore, taking it as a reference, there has been a growth in the implementation of steel framing construction systems as construction alternatives since it is seismic resistant, fast and effective [5]. In this sense, the application of a large steel section for structures such as moment-resist frames has the ability to dissipate cycling loads, prevents drifts and improves structural stiffness, however, this profile increases the

structure weight considerably [6] and the price is not always affordable, which is why it is considered as the structure to imitate by alternative materials that can be used for low cost social housing in rural areas.

The utilization of alternative materials in buildings has been researched since massive construction has led to the destruction of natural stone deposits [7], where reinforcements have been used to improve structures by increasing deformation capacity and/or its resistance [8]. Additionally, due to the need of fixing damaged structures, fiber reinforced polymers have been used in the rehabilitation of reinforced concrete structures [9]. Furthermore, the natural fibers have good mechanical properties to use as structural reinforcement in buildings [10] since fiber reinforcements enhances the strength of the base material [11], has a superior energy absorption proficiency [12] and in the design the structural engineers use materials that are determinate by market forces [13]. In this sense, fibers like coconut, sisal, jute, ramie bast, kenaf bast, abaca leaf, palm, hemp, flax, [14], cement reinforced by natural materials such as wood in particles [15], etc., are good options to be considered along with many others. In a similar way, after the Lombok earthquake of 2018, the houses that were heavily damaged reached approximately 149.706, creating the need for temporary shelter that could last, especially for people in rural areas, where timber was used, but the high demand required evaluating the implementation of other materials, where bamboo was considered for its low weight strength ratio, ductility, low cost, convenience [16] and the known fact that has a construction use [17–19].

For these cases, where there are several options, in the context of design structural elements with alternative materials is important to consider the different methodologies to make the best selection. In this sense, the designers have been helped by systematic quantitative methods who have the objective of optimizing the metrics for the best performance, looking for the maximization or minimization of an aspect [20]. This method of the performance index was developed by Prof. Mike Ashby, allowing us to rank and select materials with an optimization of coupled properties [21], and has been used in the bending strength design of beams, such as done by Burgess et al., who concluded that even though the performance index for bending strength has been derived for proportional, horizontal and vertical scaling, it also can be used in a scaling of a cross section in arbitrary directions, resulting in the explanation of why a high density material has good performance in constrained spaces [22]. Furthermore, the selection of the best material from a set of options that has a variety of aspects to consider can be a difficult task, and to solve this problem, the researchers have been using operational solutions such as the multicriteria decision methods (MCDM) [23] that have been used in the building sector to found the best materials, as done by Bisikirske et al., that employed the TOPSIS method to evaluate the best option to replace sand with glass waste in the concrete production [24]. Kiani et al. applied the VIKOR method to select the best material that allows an appropriate repair of structural concrete [25]. Furthermore, Amoozad Mahdiraji et al. considered the environmental aspects to analyze the key factors of sustainable architecture in terms of regional and cultural features with the help of the COPRAS method [26], showing that the MCDM are important tools to make the best selection in construction and materials.

However, the selection of materials needs a validation, and this can be a load simulation; in this sense, finite element analysis can be used to study structural performance since these models are inexpensive and allow us to monitor the behavior of the structures in a way that could not be easily feasible experimentally [27]. A method to simulate the behavior of beams is the discrete singular convolution (DSC) analysis, which is a numerical algorithm that provides accuracy and stability in the prediction of vibrations that occurred in beams [28]. In this sense, different authors have studied the performance of beams, like in the research of Ghabussi et al., who simulated the behavior of semi-rigid moment frames using dampers and without them, with a displacement loading protocol that increased until reaching 5% of drift ratio, proving that the damper with a 60° angle increase the frame strength and the energy dissipation [6].

For this matter, the objective of this research is to make the best selection of an alternative material for a structural beam on a social housing application. Where, the candidate materials take as reference a steel profile performance under seismic loads in Latin America, it must resist the same forces, it should be a sustainable and accessible material, which is meant to be found by software and bibliographical research. Mainly, the chosen material that represents the best among all the candidates is selected by multicriteria decision methods, and we compare its performance to the steel profile by computer-assisted simulation.

## 2. Materials and Methods

The present research considers different stages with the objective of making the selection of the best material to be used as an alternative material for beams in an application of rural housing and temporary shelter for Latin American conditions. In this sense, the present methodology explores the determination of a reference material that has been used in the region followed by a simulation of the building conditions needed for a Latin American country, using their own normative with the utilization of the software CYPECAD and Autodesk Inventor (San Rafael, CA, USA).

Once the needed conditions have been stablished, the CES-Edupack 2019 software (Canonsburg, PA, USA) is used to find the candidate materials that has to resist the before explored conditions. In this way, a list of candidate materials is evaluated, where some of them are simulated as concrete reinforcement with the synthesizer tool to use their properties for a last selection.

Moreover, the followed steps come with the utilization of the multicriteria decision methods, these techniques need a weighting of the criteria that are going to be evaluated, where the method of entropy essentially gives a better rating to the performance of the criteria that has a higher difference among the candidates [29]. On the other hand, the method of standard deviation is similar to entropy, but gives an attribute to the criteria that has similarities to the other alternatives [29]. Lastly, the preference selection index method is a weighted property method that measures the criteria according to the performance of each alternative [29]. In this sense, these three methods are used since the material alternatives should have differences that make them better or worse, and the comparative of these methods ensures that the weighting of the criteria is the nearest to accurate.

On the other hand, with the weight of the criteria, the first method considered is the method VIKOR since this method provide compromise solutions to multicriteria problems that includes conflicting criteria [30], in this case is important to consider that the criteria that are evaluated have technical, environmental and cost performance which generally are conflicting. However, the compromise solution of VIKOR sometimes results in ties, to avoid this event, a similar method is used, which is TOPSIS. This method is like VIKOR since both aim for the "closeness to ideal", but also looks to be the farthest to non-ideal [31], making it a good alternative to compare results. Lastly, in case both methods do not come to an agreement, a third method can be a tie breaker, in this case, the method COPRAS is selected since this is a technique that evaluates the significance and utility of the alternatives on conflicting decisions and has been described by the literature as simply and effective for problems that include material selection [32]. Lastly, once the best material has been selected, a validation of the winner material is performed to ensure that the material is the best.

### 2.1. Determination of the Reference Material for Beam

Taking into consideration the earth movements provoked by the oceanic Nazca plate on South America, it is important to consider a normative regarding one of these countries. The coast region of Ecuador is considered a seismic zone type VI with acceleration in reference to the gravity acceleration $\geq$50 g [33]. In the earthquake of 2016, even though the epicenter was located in Muise, finite-fault models showed that the extension of the rupture went south to Manabí, being the most damaged region since it has a soil Type C [34]. In this

case, for the construction of buildings, the normative from Ecuador (NEC-SE-VIVIENDA) is considered [3]. The normative NEC-SE-AC suggests the use of the steels ASTM A36, ASTM A572 Gr 50 and ASTM A588 Gr 50 [35]; also, following the research of Navas et al., the steel framing construction system has been used for the decks in the country with the material ASTM A572 [5].

### 2.2. Load Determination

The determination of the distributed load that the beams must endure is found by the utilization of the software CYPECAD with the following steps:

1. Normative for annealed steel: The rule NEC-SE-AC make reference to the norm ANSI/AISC 341-10 and ANSI/AISC 360-10 [35]. The second one is selected.
2. Profiles of laminated steel: Navas et al. determined the use of the steel ASTM A572 with an ultimate tensile strength of 60 ksi, so this is selected.
3. Seismic Action: For Ecuador NEC-SE-DS-2014:
   - Analysis method—Dynamic;
   - Spectrum definition—Norm;
   - Seismic zone—VI [33];
   - Region—Coast [34];
   - Soil—Type C [34].
4. Structural design of columns: According to Figure 1 with a load (CM) of 0.7 KN·m$^{-2}$ [33] and a height of 2 m.
5. Structural design of beams: Navas et al. used a Type G profile; however, the software only has the Type C with the smallest measures of a beam UPN50. This type of beam has been spotted to be used for buildings in the research of Castañeda and Mieles Bravo [4].
6. The celling is chosen as a default slab.
7. Calculations without foundation: Metallic beams using all series profiles.

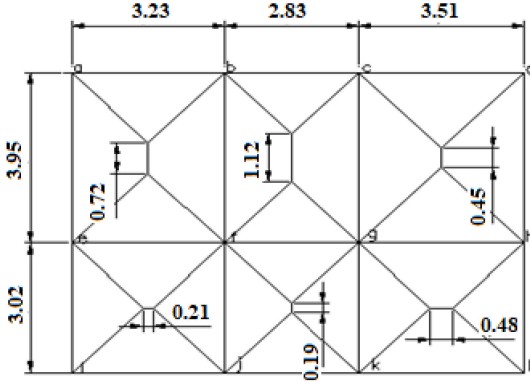

**Figure 1.** House design.

### 2.3. Steel Beam Yield Strength

Once the simulation of the earthquake has thrown the loads that the beams must resist, a stress analysis simulation with the Autodesk Inventor software is performed with the design of the CAD shown in Figure 2. The frame analysis was utilized, since it uses a DSC algorithm that generates a node in the structure allowing us to define elastic releases in the beam [36]. In this sense, the joints can be simulated as fixed nodes in the extremes of the profile, the loads can be applied in this joint of the beam with the magnitude and direction found in the CYPE simulation for the worst case and it includes the gravity as displayed in Figure 3, allowing us to know the strength that the alternative material must endure.

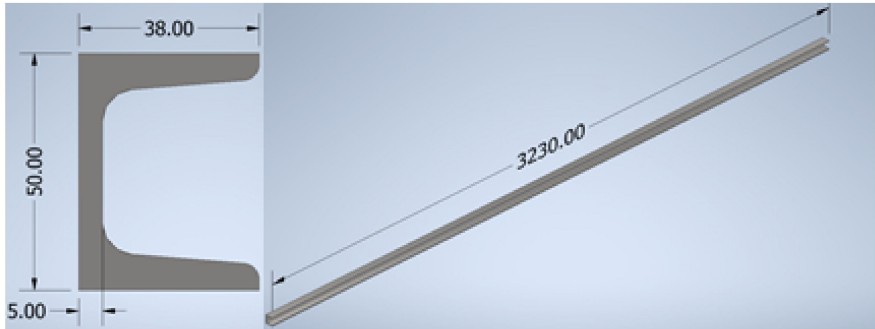

**Figure 2.** Beam model.

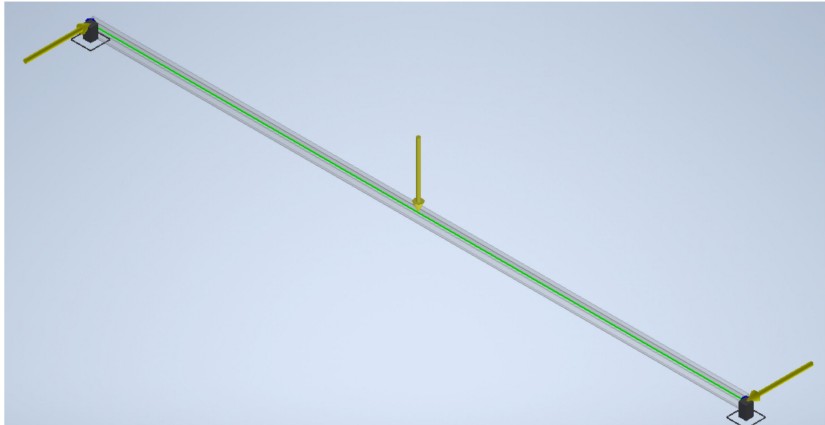

**Figure 3.** Loads display.

*2.4. Selection by CES-EDUPACK*

First, by using the software specialized in material selection CES-EDUPACK, with the level 3 Eco-Design library, the search for the best material begins, and in this sense, the following steps are taken with the guide of the software CES-EDUPACK (Canonsburg, PA, USA) by Granta-Design [21]:

1. Tree Hierarchy: A tree is displayed with all the records in the library data base and grouped by hierarchy, allowing us to filter the records. In this stage, the groups of "natural fibers", "hybrids" and "polymers" are selected. Additionally, since the material ASTM A572 is going to be replaced, it has to be considered, however, this is not available so, taking into consideration the chemical composition and the mechanical requests expressed in the Ecuadorian technical normative for hot rolled steel profiles [37], the low carbon steels for general construction applications as rolled AISI 1015 and AISI 1022 are also included [21].

2. Limit: Stage 2 limits the materials by technical properties, here a minimum yield strength of 5.96 MPa is defined since this is the load that the C profile managed in the simulation. Since seismic events are being considered, a minimum limit in Poisson's ratio of 0.29 is considered, this being equal to the steel in the profile. Furthermore using the website of CYPE, the material ASTM A572 in Ecuador has a reference price of 1.88 USD·kg$^{-1}$ [38], which is used as the reference maximum limit. Additionally, is important that the replace material does not provoke a greater environmental impact, by this means, the embodied energy and $CO_2$ footprints of the primary production of the steels are considered as maximum limits being 33.9 MJ·kg$^{-1}$ and 2.49 $CO_2$ kg·kg$^{-1}$, respectively.

3. Chart: In this stage, the materials are evaluated for the performance indexes with two different comparisons.

- First, since seismic movements are considered, the evaluation is done in terms of flexural vibration for the maximum resonant frequency expressed by $E_f^{1/2} x \rho^{-1}$ vs. the minimum vibration amplitude determined by $\rho x \left( \eta x E_f^{1/2} \right)^{-1}$, where $E_f$ is the flexural modulus, $\rho$ the density and $\eta$ the mechanical loss coefficient (damping coefficient).
- To optimize the chart, an index line is included, taking as reference the bamboo since it has been studied as an alternative material, showing to be a good construction material and good for resisting earthquakes [19].

4.  Final selection: The remaining materials were considered as the most suitable; however, some of them are repeated and are only differentiated for the disposition of the particle or fiber, in this sense, when a particle is perpendicular to the force applied, it is stronger, being more resistant to the parallel disposition for columns in compression [15], hence for beams, the perpendicular arrays are selected as well as the materials that have been proven to be useful for buildings. Furthermore, other materials are not useful as themselves, but they can be used as composites for reinforced concrete (RC) in beams.

In this sense, following the aforementioned steps, a list of candidate materials was obtained to analyze which is the best material to be used as a beam in the application of rural housing and temporary shelter for Latin American conditions.

### 2.5. Synthesizer Tool

The selection of the candidate materials left the option of alternatives that can be useful only as RC; however, these materials are not part of the library. In this sense, to know their potential properties, the CES-EDUPACK Software allows us to use the synthesizer tool to create such materials. For this case, the ordinary Portland cement was used with the reinforce of short fibers [14] of coir, flax, palm 0.5, polypropylene and sisal. The factors to be considered were aligned orientation, volume fraction from 5% to 50% and number of values as fiber aspect ratio default. Likewise, the results are evaluated with a chart method. Finally, with this, the list of candidate materials is complete.

### 2.6. Multicriteria Decision Methods

The multicriteria decision methods are operational tools that allows us to make the best choice on the field of material selection. To do so, the candidate material has been chosen by its performance on technical, environmental, and cost aspects. These numerical values obtained by the software CES-EDUPACK 2019 are used to assess the options by MCDM means.

#### 2.6.1. Entropy Method

Following the method expressed by Zhu et al., a decision matrix is developed from the criteria of every material Equation (1), followed by the calculation of the normalized matrix and the entropy calculation Equation (2) and lastly the weight of each criteria Equation (3) [39]. The equations for this method are used as follows:

$$P_{ij} = \frac{x_{ij}}{\sum_{j=1}^n x_{ij}} \tag{1}$$

$$E_i = \frac{\sum_{j=1}^n P_{ij} \times \ln P_{ij}}{\ln n} \tag{2}$$

$$w_i = \frac{1 - E_i}{\sum_{i=1}^m (1 - E_i)} \tag{3}$$

### 2.6.2. Standard Deviation Method

In the research of Jahan et al., the author expresses the steps to follow this weight method by the calculation of disperse values in function of the average Equation (4) and then the weight of each criteria is obtained Equation (5) [29] with the corresponding equations that follow:

$$\sigma_j = \sqrt{\frac{\sum_{i=1}^m \left(x_{ij} - \overline{x_j}\right)^2}{m}} \tag{4}$$

$$w_j = \frac{\sigma_j}{\sum_{j=1}^n \sigma_j} \tag{5}$$

### 2.6.3. Preference Selection Index Method

The preference selection index method for weight criteria is described in the research of Jahan et al., where the calculations start with the determination of minimum and maximum criteria, followed by a normalized matrix that uses the determination of beneficial (higher is better) Equation (6) and non-beneficial (higher is worst/cost) criteria Equation (7), which is described in Table 1. Then, the calculation continues with the preference variation value Equation (8) divergence level of the diversity of values Equation (10) and finally the weight of every criteria Equation (11) [29] with the following equations:

$$r_{ij} = \frac{x_{ij}}{x_j^{max}} \tag{6}$$

$$r_{ij} = \frac{x_j^{min}}{x_{ij}} \tag{7}$$

$$PV_j = \sum_{i=1}^m \left(r_{ij} - \overline{r_j}\right)^2 \tag{8}$$

$$\overline{r_j} = \frac{\sum_{i=1}^m r_{ij}}{m} \tag{9}$$

$$\varphi_j = 1 - PV_j \tag{10}$$

$$w_j = \frac{\varphi_j}{\sum_{j=1}^n \varphi_j} \tag{11}$$

**Table 1.** Criteria benefit classification.

| Criteria | Classification |
|---|---|
| Price | Non-Beneficial |
| Density | Non-Beneficial |
| Young's Modulus | Beneficial |
| Yield Strength | Beneficial |
| Flexural Modulus | Beneficial |
| Embodied Energy | Non-Beneficial |
| $CO_2$ Footprint | Non-Beneficial |

### 2.6.4. VIKOR Method

The ranking of the VIKOR method was developed with the method of Opricovic and Tzeng, following the steps of determination of maximum and minimum values using the criteria classification of Table 1, then the calculations of the values of utility $S_i$ Equation (12) and regret $R_i$ Equation (13), followed by the calculation of the ranking with the value $Q_i$ Equation (14), using the factor $v = 0.5$ of the maximum group of utility. Furthermore, the

compromise solutions with its acceptable advantage Equation (15) and acceptable stability are also assessed [31] by using the next equations:

$$S_i = \sum_{j=1}^{n} W_j \frac{X_j^* - X_{ij}}{X_j^* - X_j^-} \tag{12}$$

$$R_i = \max_j W_j \frac{X_j^* - X_{ij}}{X_j^* - X_j^-} \tag{13}$$

$$Qi = v * \frac{S_i - S^*}{S^- - S^*} + (1 - v) * \frac{R_i - R^*}{R^- - R^*} \tag{14}$$

$$QA^{(2)} - QA^{(1)} \geq DQ \tag{15}$$

where:

$$DQ = \frac{1}{n-1} \tag{16}$$

### 2.6.5. TOPSIS Method

The selection of the best material by the TOPSIS method is also made with the methodology of Papathanasiou and Ploskas, calculating the normalization of the decision matrix for TOPSIS using the technique of vector normalization, Equation (17), this is weighted for the values of each criteria, Equation (18), then comes the determination of the ideal and anti-ideal solutions using the determinations of Table 1, followed by the separation distances to ideal solution, Equation (19), and anti-ideal solution, Equation (20), followed by the closeness to the ideal solution, Equation (21), that allows to rank the materials [40].

$$r_{ij} = \frac{X_{ij}}{\sqrt{\sum_{j=1}^{n} X_{ij}^2}} \tag{17}$$

$$V_{ij} = r_{ij} * W_j \tag{18}$$

$$D_j^* = \left(\sum_{j=1}^{m} (V_{ij} - V^+)^2\right)^{0.5} \tag{19}$$

$$D_j^- = \left(\sum_{j=1}^{m} (V_{ij} - V^-)^2\right)^{0.5} \tag{20}$$

$$C_j = \frac{D_j^-}{D_j^* + D_j^-} \tag{21}$$

### 2.6.6. COPRAS Method

The ranking of the COPRAS method is reached with following steps taken from Mousavi-Nasab and Sotoudeh-Anvari, first the normalization matrix, Equation (22), and then the weighted matrix, Equation (23), followed by the obtention of the weighted normalized scores, Equations (24) and (25), according to the classification expressed in Table 1. Lastly, the calculation of comparative significance $Q_i$ Equation (26), that allows to find the level of choice utility, Equation (27), that gives us the final ranking of the method [41].

$$r_{ij} = \frac{X_{ij}}{\sum_{i=1}^{m} X_{ij}} \tag{22}$$

$$y_{ij} = r_{ij} * W_j \tag{23}$$

$$S_{+i} = \sum_{j=1}^{n} y_{+ij} \tag{24}$$

$$S_{-i} = \sum_{j=1}^{n} y_{-ij} \tag{25}$$

$$Q_i = S_{+i} + \frac{\sum_{i=1}^{m} S_{-i}}{S_{-i} \sum_{i=1}^{m} (S_{-i})^{-1}} \tag{26}$$

$$U_i = \left(\frac{Q_i}{Q_{max}}\right) * 100\% \tag{27}$$

### 2.6.7. Material Simulation

In order to validate the best material selected according to the MCDM, a simulation of a beam is performed by finite element analysis. In this sense, the best material considered is the bamboo; however, according to the literature, the diameter of the bamboo specie Guadua in Latin America presents a diameter between 120 mm to 160 mm [40] and the width along the tangential direction can be from 5 mm to 20 mm [41]. In this sense, the model will consider the lowest diameter and the minimum width, as it can be seen in Figure 4. Additionally, the same loads applied before for the steel beam are applied here, as it can be seen in Figure 5.

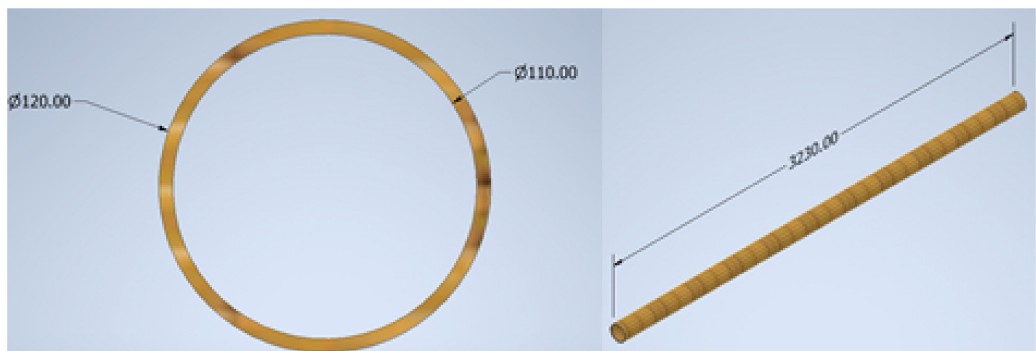

**Figure 4.** Bamboo beam model.

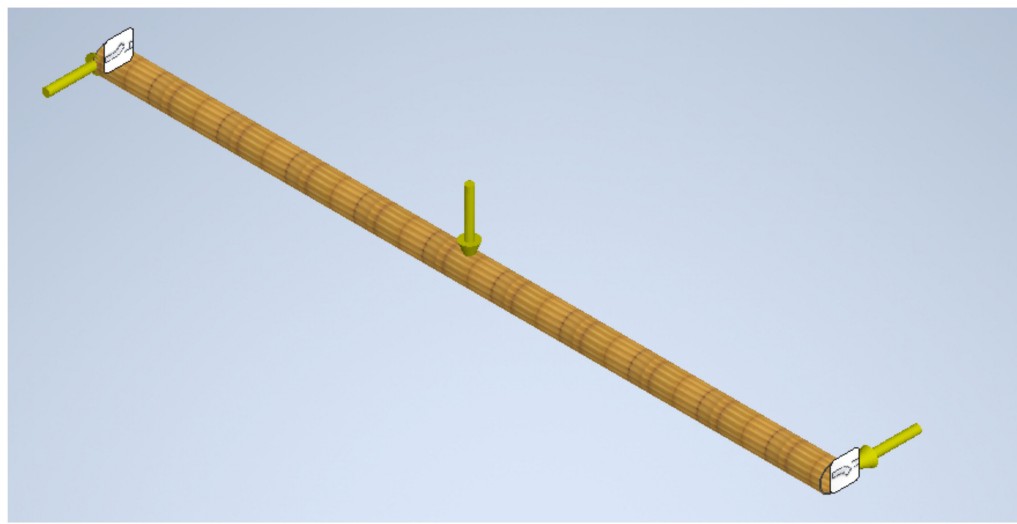

**Figure 5.** Bamboo beam loads.

## 3. Results and Discussion

### 3.1. Reference Loads Simulations

After the modelling of the structure, it is shown in Figure 6. Furthermore, the simulation of the loads resulting from the wraps shows that the joints are the zone that have

the highest loads; in this sense, it has been identified that the column-beam joints are the most common cause of failure in moment-resist frames due to the shear failure [9], as it can be seen in Figure 7 for the highest load, and Table 2 displays all the values, where the negative load of C6 represents the highest load with 12.17 KN. Furthermore, the result of the simulation performed in the steel profile is presented in Figure 8 where the maximum normal stress is 5.96 MPa located in the joint of the beam and the column.

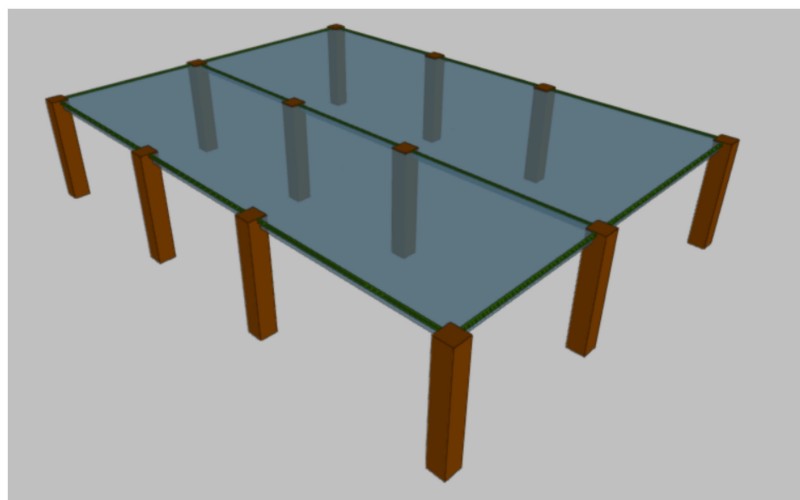

**Figure 6.** Structure model.

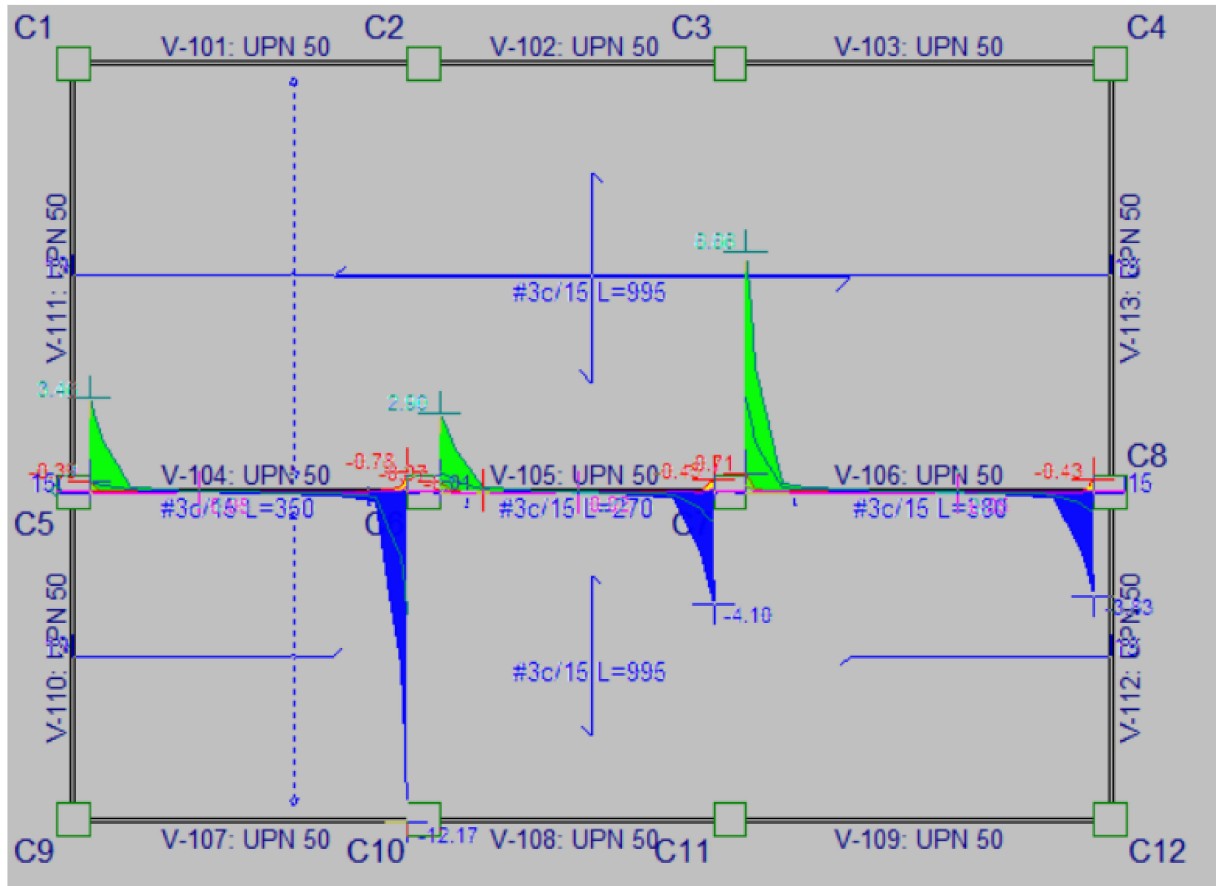

**Figure 7.** Highest seismic load simulation.

**Table 2.** Simulation loads.

| Beam | Load (Q+) KN | Load (Q−) KN |
|:---:|:---:|:---:|
| C1–C2 | 3.13 | 6.98 |
| C2–C3 | 1.31/1.76 | 1.47/1.92 |
| C3–C4 | 5.95 | 3.85 |
| C5–C6 | 3.46 | 12.17 |
| C6–C7 | 2.90 | 4.10 |
| C7–C8 | 8.86 | 3.83 |
| C9–C10 | 3.31 | 6.58 |
| C10–C11 | 1.89/0.99 | 0.7/2.57 |
| C11–C12 | 6.17 | 4.01 |
| C1–C5 | 5.17 | 4.02 |
| C5–C9 | 2.44 | 2.35 |
| C4–C8 | 5.45 | 3.97 |
| C8–C12 | 2.59 | 2.33 |

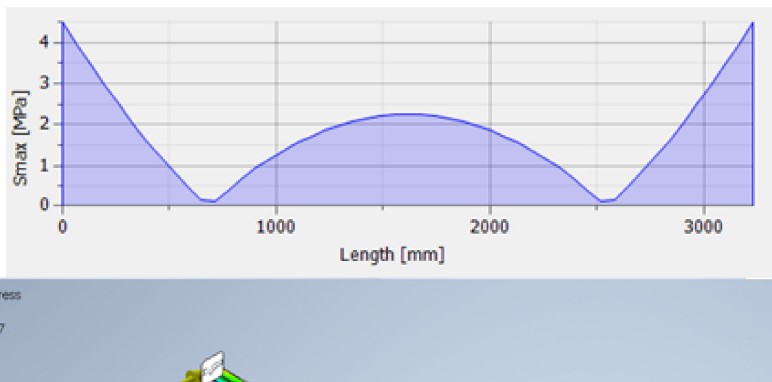

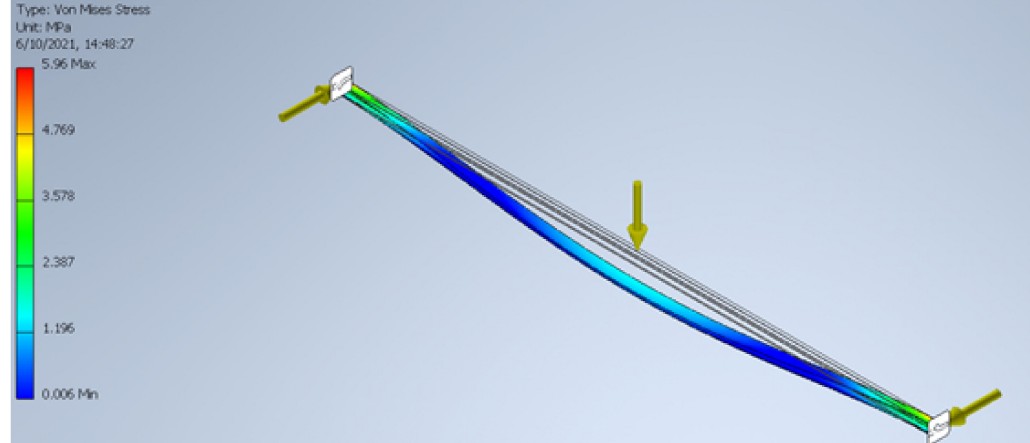

**Figure 8.** Beam load result.

Here, it can be seen that the joints are the zones that suffers most of the loads in opposite directions, an issue that has been seen in different studies such as the performed by Saber and Shahria, where steel beams-columns with post-tensioned elements that eliminate deformations on moment-resisting frames for seismic events have been simulated; here, the simulation and the experimental results show a yield in the angles of the joint, a gap between the beam and the column and the deformations in the local bucking in the adjustments of the profile, all due to the forces that the joints have to endure due to the seismic events [27].

*3.2. CES-EDUPACK Material Selection*

The utilization of the software CES-EDUPACK allowed us to reduce the number of candidate materials as expressed in Table 3, where the third stage of the selection displays the performance index chart that can be found in Figure 9. Furthermore, the list of candidates that has been proven to be useful in building applications is shown in Table 4,

where the materials are sorted in alphabetical order and do not reflect any preference. In this sense, it is feasible to consider fibers that are reinforcement of concrete as expressed in the methodology, since, over the years, different kinds of fibers have been used by researchers as construction materials due to the price, local availability and improving properties [14].

**Table 3.** CES-EDUPACK Selection.

| Stage | Criteria | Quantity of Materials |
|---|---|---|
| Initial | Level 3 Eco-Design Library | 4026 |
| 1st | Tree Hierarchy | 1872 |
| 2nd | Limit: Yield Strength/Price/Energy/$CO_2$ | 194 |
| 3rd | Chart: Performance index. Figure 9 | 26 |
| 4th | Selection: Perpendicular dispositions and proven building applications | 7 |

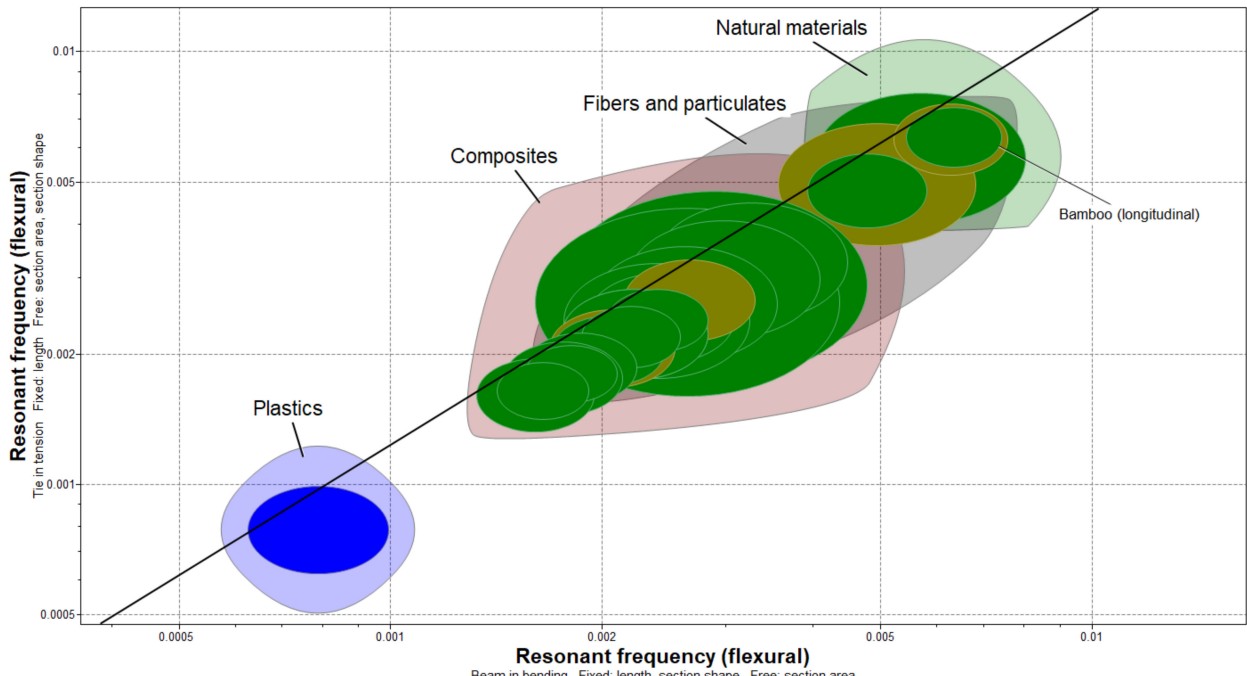

**Figure 9.** Performance index chart.

**Table 4.** Candidate beam material.

| Material | Reference for Building |
|---|---|
| Bamboo | [19] |
| Cement Bonded Particle Board, perpendicular to board | [15] |
| Coir (RC) | [14] |
| Flax (RC) | [10] |
| Palm 0.50 (RC) | [7] |
| Polypropylene (RC) | [11] |
| Sisal (RC) | [12] |

*3.3. Synthesizer Results*

After the development of different simulations of reinforced concrete with fibers, six results for each composite are obtained, and to select the best a comparative chart of yield strength vs. flexural modulus is created. In this way, Figure 10 shows the distribution, and the best are highlighted considering higher yield strength as more important since

it has been studied that, in the seismic design, the yield displacement is a natural and useful parameter to consider [13]. Here, the composites of ordinary cement and a volume fraction of 41.9% coir; 50% sisal; 50% flax; 50% polypropylene; and 50% palm 0.5 are chosen. Furthermore, Table 5 displays the properties of all candidate materials with the criteria that will be assessed, also, an M index is included to make them easier to organize.

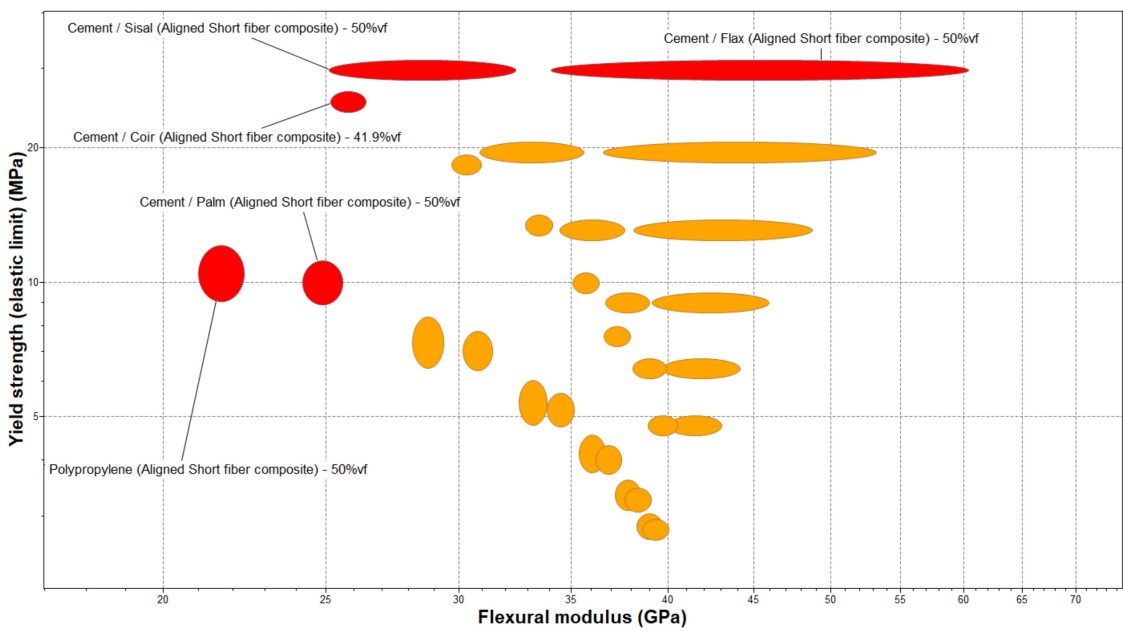

**Figure 10.** Yield strength vs. flexural modulus on composites.

**Table 5.** Candidate materials.

| Material | Index | Price (USD·kg$^{-1}$) | Density (kg·m$^{-3}$) | Young's Modulus (GPa) | Yield Strength (MPa) | Flexural Modulus (GPa) | Embodied Energy (MJ·kg$^{-1}$) | CO$_2$ Footprint (kg·kg$^{-1}$) |
|---|---|---|---|---|---|---|---|---|
| Bamboo | M1 | 2.01 | 800 | 15 | 35.9 | 17 | 35.9 | 1.16 |
| Cement Bonded Particle Board | M2 | 0.27 | 1300 | 1.8 | 3.6 | 3 | 8.87 | 0.66 |
| Coir (RC) | M3 | 0.21 | 1790 | 25.2 | 24.1 | 25.2 | 5.54 | 0.88 |
| Flax (RC) | M4 | 1.25 | 1860 | 34.1 | 28.4 | 34.1 | 6.97 | 0.49 |
| Palm 0.5 (RC) | M5 | 0.427 | 1460 | 24.2 | 8.95 | 24.2 | 6.02 | 0.569 |
| Polypropylene (RC) | M6 | 0.77 | 2110 | 21 | 9.06 | 21 | 18.3 | 0.857 |
| Sisal (RC) | M7 | 0.579 | 1830 | 25.1 | 28.4 | 25.1 | 6.37 | 1.02 |

In this way, the eight materials that have been chosen as candidates to be used in a beam application with their corresponding mechanical, environmental and price characteristics, now pass to the final evaluation of the best material by the utilization of multicriteria decision methods.

### 3.4. Entropy Results

The calculations made with formulas in the research of Zhu et al., are presented in Table 6 with a normalized matrix, and Table 7 with the entropy and weight of the criteria.

**Table 6.** Normalized matrix entropy.

| Normalized Matrix $P_{ij}$ | | | | | | |
|---|---|---|---|---|---|---|
| 0.36 | 0.07 | 0.10 | 0.26 | 0.11 | 0.41 | 0.21 |
| 0.05 | 0.12 | 0.01 | 0.03 | 0.02 | 0.10 | 0.12 |
| 0.04 | 0.16 | 0.17 | 0.17 | 0.17 | 0.06 | 0.16 |
| 0.23 | 0.17 | 0.23 | 0.21 | 0.23 | 0.08 | 0.09 |
| 0.08 | 0.13 | 0.17 | 0.06 | 0.16 | 0.07 | 0.10 |
| 0.14 | 0.19 | 0.14 | 0.07 | 0.14 | 0.21 | 0.15 |
| 0.10 | 0.16 | 0.17 | 0.21 | 0.17 | 0.07 | 0.18 |

**Table 7.** Entropy weight.

| Criteria | Entropy | Weight | Rank |
|---|---|---|---|
| Price | 0.87 | 0.25 | 2 |
| Density | 0.98 | 0.03 | 7 |
| Young's Modulus | 0.93 | 0.13 | 4 |
| Yield Strength | 0.90 | 0.18 | 3 |
| Flexural Modulus | 0.94 | 0.11 | 5 |
| Embodied Energy | 0.86 | 0.26 | 1 |
| $CO_2$ Footprint | 0.98 | 0.04 | 6 |

The results show that for this weight method, the most important criteria are the embodied energy, followed by the price and the yield strength, in this case it would be useful when prioritizing the environment, but also considering the economical need of implementing a cheap resisting beam.

### 3.5. Standard Deviation Results

In Table 8, the standard deviation of the criteria and the weights are a calculated, along is presented the ranking of the weights where the density is considered the most important with the yield strength and the embodied energy. In this sense, the method could be use when the construction and resistance to bending aspects are needed, followed by an environmental consideration.

**Table 8.** Standard deviation weight result.

| Criteria | Standard Deviation | Weight | Rank |
|---|---|---|---|
| Price | 0.60 | 0.0013 | 6 |
| Density | 408.12 | 0.9087 | 1 |
| Young's Modulus | 9.42 | 0.0210 | 4 |
| Yield Strength | 11.47 | 0.0255 | 2 |
| Flexural Modulus | 8.91 | 0.0198 | 5 |
| Embodied Energy | 10.37 | 0.0231 | 3 |
| $CO_2$ Footprint | 0.23 | 0.0005 | 7 |

### 3.6. Preference Selection Index Result

From the method of preference selection, Table 9 displays the normalized matrix, and in Table 10, the results of divergence, weight, and ranking. In this case, the results present a mixture between the before mentioned methods. Where an environmental aspect is considered as the most important followed by the density and lastly the flexural modulus.

**Table 9.** Normalized matrix for Preference Selection Index.

| Price (USD·kg$^{-1}$) | Density (kg·m$^{-3}$) | Young's Modulus (GPa) | Yield Strength (MPa) | Flexural Modulus (GPa) | Embodied Energy (MJ·kg$^{-1}$) | CO$_2$ Footprint (kg·kg$^{-1}$) |
|---|---|---|---|---|---|---|
| 0.12 | 0.20 | 0.03 | 0.20 | 0.02 | 0.26 | 0.06 |
| 0.10 | 0.00 | 0.31 | 0.20 | 0.29 | 0.00 | 0.01 |
| 0.30 | 0.01 | 0.02 | 0.01 | 0.01 | 0.11 | 0.01 |
| 0.08 | 0.01 | 0.15 | 0.06 | 0.14 | 0.02 | 0.11 |
| 0.00 | 0.00 | 0.01 | 0.09 | 0.01 | 0.06 | 0.04 |
| 0.03 | 0.03 | 0.00 | 0.09 | 0.00 | 0.13 | 0.01 |
| 0.01 | 0.01 | 0.02 | 0.06 | 0.01 | 0.04 | 0.03 |

**Table 10.** PSI weight result.

| Criteria | Divergence | Weight | Rank |
|---|---|---|---|
| Price | 0.35 | 0.10 | 6 |
| Density | 0.73 | 0.21 | 2 |
| Young's Modulus | 0.47 | 0.14 | 4 |
| Yield Strength | 0.29 | 0.08 | 7 |
| Flexural Modulus | 0.52 | 0.15 | 3 |
| Embodied Energy | 0.37 | 0.11 | 5 |
| CO$_2$ Footprint | 0.73 | 0.21 | 1 |

Since the objective of this research is to find an alternative material that can resist seismic events, it is considered that the priority has to be laid in the mechanical properties, for this reason among the three methods, the standard deviation weights were selected as the best suited to be used in the following multicriteria decision methods.

*3.7. VIKOR Results*

In the method VIKOR Table 11, shows the calculations of the maximum and minimum values with each element of the decision matrix and the weight from the standard deviation method, these values allow to calculate the terms of $S_i$, $R_i$ and $Q_i$ in Table 12, resulting in a ranking of the materials. The best material is the M1 corresponding to the bamboo.

**Table 11.** VIKOR calculations.

| Price (USD·kg$^{-1}$) | Density (kg·m$^{-3}$) | Young's Modulus (GPa) | Yield Strength (MPa) | Flexural Modulus (GPa) | Embodied Energy (MJ·kg$^{-1}$) | CO$_2$ Footprint (kg·kg$^{-1}$) |
|---|---|---|---|---|---|---|
| 0.00133 | 0.00000 | 0.01240 | 0.00000 | 0.01090 | 0.02309 | 0.00050 |
| 0.00004 | 0.34684 | 0.02097 | 0.02554 | 0.01983 | 0.00253 | 0.00013 |
| 0.00000 | 0.68675 | 0.00578 | 0.00933 | 0.00568 | 0.00000 | 0.00029 |
| 0.00077 | 0.73531 | 0.00000 | 0.00593 | 0.00000 | 0.00109 | 0.00000 |
| 0.00016 | 0.45784 | 0.00643 | 0.02131 | 0.00631 | 0.00037 | 0.00006 |
| 0.00041 | 0.90873 | 0.00850 | 0.02122 | 0.00835 | 0.00971 | 0.00028 |
| 0.00027 | 0.71450 | 0.00584 | 0.00593 | 0.00574 | 0.00063 | 0.00040 |

Furthermore, for the calculations of the compromise solution alternatives, the first condition of acceptable advantage of $Q(a'') - Q(a') \geq DQ$ is accomplished since the factor $DQ$ for seven candidates is 0.17 and the difference between the first and the second option is 0.38. Additionally, the values of $S_i$ and $R_i$ are the nearest to cero for the M1 in both cases making it to fulfil the condition of acceptable stability.

**Table 12.** VIKOR ranking.

| Material | $S_i$ | $R_i$ | $Q_i$ | Rank |
|---|---|---|---|---|
| Bamboo | 0.05 | 0.02 | 0.00 | 1 |
| Cement Bonded Particle Board | 0.42 | 0.35 | 0.39 | 2 |
| Coir (RC) | 0.71 | 0.69 | 0.74 | 4 |
| Flax (RC) | 0.74 | 0.74 | 0.78 | 6 |
| Palm 0.5 (RC) | 0.49 | 0.46 | 0.49 | 3 |
| Polypropylene (RC) | 0.96 | 0.91 | 1.00 | 7 |
| Sisal (RC) | 0.73 | 0.71 | 0.77 | 5 |

*3.8. TOPSIS Results*

Starting with the normalization, the results of the equations expressed by Papathanasiou and Ploskas are displayed in Table 13, then the weighted matrix is represented in Table 14 and in Table 15 the separation measures ($D_i$), relative closeness ($C_i$) and the ranking of the method. In this case, again the method choses the bamboo material as the best.

**Table 13.** Normalized TOPSIS matrix.

| Price (USD·kg$^{-1}$) | Density (kg·m$^{-3}$) | Young's Modulus (GPa) | Yield Strength (MPa) | Flexural Modulus (GPa) | Embodied Energy (MJ·kg$^{-1}$) | $CO_2$ Footprint (kg·kg$^{-1}$) |
|---|---|---|---|---|---|---|
| 0.77 | 0.18 | 0.25 | 0.59 | 0.28 | 0.83 | 0.52 |
| 0.10 | 0.30 | 0.03 | 0.06 | 0.05 | 0.21 | 0.30 |
| 0.08 | 0.41 | 0.42 | 0.40 | 0.41 | 0.13 | 0.40 |
| 0.48 | 0.43 | 0.56 | 0.47 | 0.56 | 0.16 | 0.22 |
| 0.16 | 0.34 | 0.40 | 0.15 | 0.40 | 0.14 | 0.26 |
| 0.29 | 0.49 | 0.35 | 0.15 | 0.34 | 0.42 | 0.39 |
| 0.22 | 0.42 | 0.41 | 0.47 | 0.41 | 0.15 | 0.46 |

**Table 14.** TOPSIS weighted matrix.

| Price (USD·kg$^{-1}$) | Density (kg·m$^{-3}$) | Young's Modulus (GPa) | Yield Strength (MPa) | Flexural Modulus (GPa) | Embodied Energy (MJ·kg$^{-1}$) | $CO_2$ Footprint (kg·kg$^{-1}$) |
|---|---|---|---|---|---|---|
| 0.0010 | 0.1671 | 0.0052 | 0.0152 | 0.0055 | 0.0192 | 0.0003 |
| 0.0001 | 0.2715 | 0.0006 | 0.0015 | 0.0010 | 0.0048 | 0.0001 |
| 0.0001 | 0.3739 | 0.0087 | 0.0102 | 0.0082 | 0.0030 | 0.0002 |
| 0.0006 | 0.3885 | 0.0118 | 0.0120 | 0.0110 | 0.0037 | 0.0001 |
| 0.0002 | 0.3050 | 0.0084 | 0.0038 | 0.0078 | 0.0032 | 0.0001 |
| 0.0004 | 0.4407 | 0.0073 | 0.0038 | 0.0068 | 0.0098 | 0.0002 |
| 0.0003 | 0.3823 | 0.0087 | 0.0120 | 0.0081 | 0.0034 | 0.0002 |

**Table 15.** TOPSIS ranking.

| Material | Separation to Ideal ($D_i$*) | Separation to Anti-Ideal ($D_i^-$) | Closeness to Ideal ($C_i$) | Rank |
|---|---|---|---|---|
| Bamboo | 0.018 | 0.27 | 0.94 | 1 |
| Cement Bonded Particle Board | 0.106 | 0.17 | 0.61 | 2 |
| Coir (RC) | 0.207 | 0.07 | 0.25 | 4 |
| Flax (RC) | 0.221 | 0.06 | 0.21 | 6 |
| Palm 0.5 (RC) | 0.138 | 0.14 | 0.50 | 3 |
| Polypropylene (RC) | 0.274 | 0.01 | 0.05 | 7 |
| Sisal (RC) | 0.215 | 0.06 | 0.22 | 5 |

*3.9. COPRAS Results*

The results of the calculation of the COPRAs method are shown in the following tables, with Table 16 for the normalized matrix, Table 17 for the weighted matrix and the weighted

normalized scores, comparative significance, level of utility and ranking in Table 18, which reaffirms the bamboo as the most suitable selection.

**Table 16.** COPRAS normalized matrix.

| Price (USD·kg$^{-1}$) | Density (kg·m$^{-3}$) | Young's Modulus (GPa) | Yield Strength (MPa) | Flexural Modulus (GPa) | Embodied Energy (MJ·kg$^{-1}$) | CO$_2$ Footprint (kg·kg$^{-1}$) |
|---|---|---|---|---|---|---|
| 0.36 | 0.07 | 0.10 | 0.26 | 0.11 | 0.41 | 0.21 |
| 0.05 | 0.12 | 0.01 | 0.03 | 0.02 | 0.10 | 0.12 |
| 0.04 | 0.16 | 0.17 | 0.17 | 0.17 | 0.06 | 0.16 |
| 0.23 | 0.17 | 0.23 | 0.21 | 0.23 | 0.08 | 0.09 |
| 0.08 | 0.13 | 0.17 | 0.06 | 0.16 | 0.07 | 0.10 |
| 0.14 | 0.19 | 0.14 | 0.07 | 0.14 | 0.21 | 0.15 |
| 0.10 | 0.16 | 0.17 | 0.21 | 0.17 | 0.07 | 0.18 |

**Table 17.** COPRAS weighted matrix.

| Price (USD·kg$^{-1}$) | Density (kg·m$^{-3}$) | Young's Modulus (GPa) | Yield Strength (MPa) | Flexural Modulus (GPa) | Embodied Energy (MJ·kg$^{-1}$) | CO$_2$ Footprint (kg·kg$^{-1}$) |
|---|---|---|---|---|---|---|
| 0.00056 | 0.04395 | 0.00146 | 0.03840 | 0.00369 | 0.03097 | 0.00064 |
| 0.00007 | 0.07142 | 0.00018 | 0.00385 | 0.00065 | 0.00765 | 0.00037 |
| 0.00006 | 0.09833 | 0.00245 | 0.02578 | 0.00547 | 0.00478 | 0.00049 |
| 0.00035 | 0.10218 | 0.00332 | 0.03038 | 0.00740 | 0.00601 | 0.00027 |
| 0.00012 | 0.08020 | 0.00236 | 0.00957 | 0.00525 | 0.00519 | 0.00031 |
| 0.00021 | 0.11591 | 0.00204 | 0.00969 | 0.00456 | 0.01579 | 0.00047 |
| 0.00016 | 0.10053 | 0.00244 | 0.03038 | 0.00545 | 0.00550 | 0.00056 |

**Table 18.** COPRAS ranking.

| Material | $S + i$ | $S - i$ | $Q_i$ | $U_i$ | Rank |
|---|---|---|---|---|---|
| Bamboo | 0.04 | 0.08 | 0.17 | 100% | 1 |
| Cement Bonded Particle Board | 0.00 | 0.08 | 0.12 | 74% | 6 |
| Coir (RC) | 0.03 | 0.10 | 0.13 | 74% | 5 |
| Flax (RC) | 0.04 | 0.11 | 0.13 | 76% | 2 |
| Palm 0.5 (RC) | 0.02 | 0.09 | 0.13 | 76% | 3 |
| Polypropylene (RC) | 0.02 | 0.13 | 0.09 | 52% | 7 |
| Sisal (RC) | 0.04 | 0.11 | 0.13 | 76% | 4 |

In this way, the utilization of the weighting method of standard deviation allowed us to assess the criteria needed to make an adequate decision that fulfils the needs of a beam for rural housing and temporary shelter on Latin American conditions. Furthermore, the application of the multicriteria decision methods VIKOR, TOPSIS and COPRAS comes to an agreement, indicating that the best material among all is the bamboo.

This material has important characteristics in the technical and environmental aspects that makes it the best option according to the MCDM. Furthermore, the work of Benton about the species of bamboo express that in a distribution from Mexico to Argentina the specie of bamboo that can be found is the Gudua Angustfolia, that has applications in construction being an excellent material for the task [17]. The selection by multicriteria decision methods has been used in construction problems and has been proven to be a powerful tool in decision making all over the world helping the building sector to evolve in the consideration of conflicting criteria [23], in fact, by the utilization of MCDM the research of Formisano et al., proved that the buckling restrained braces were the best retrofitting system for a seismic upgrading for a building [8], which gives us the security that the selection of the bamboo by MCDM is the best option. Furthermore, an advantage of the

bamboo is that there are over 1250 species around the world [17], and it can be used from the first 4 to 6 months of life since it develops its full height, diameter and thickness [18], making it feasible to found, is sustainable and can be used in developing countries such as India [19] or Latin American countries.

### 3.10. Bamboo Simulation

The simulation was performed with the technical characteristics provided by the software CES-EDUPACK and involving loads simulated in the CYPE CAD, and the beam of bamboo presented a great resistance and even superior to the profile that has been in use, as can be seen in Figure 11, and Figure 12 shows that the use of Guadua has a maximum displacement of 0.069 mm, which is very low and makes the material safe in every instance for these conditions, something that is represented in Figure 13 by the safety factor simulation. In this sense, the work of Archila-Santos et al. reported that in the year 1999, the coffee growing region of Colombia suffered a devastating earthquake of a magnitude of 6.2 in the Richter Scale, where most of the buildings made of Guadua survived the event with minor damage when compared with the structures made of concrete, of which almost 60% fell down [18].

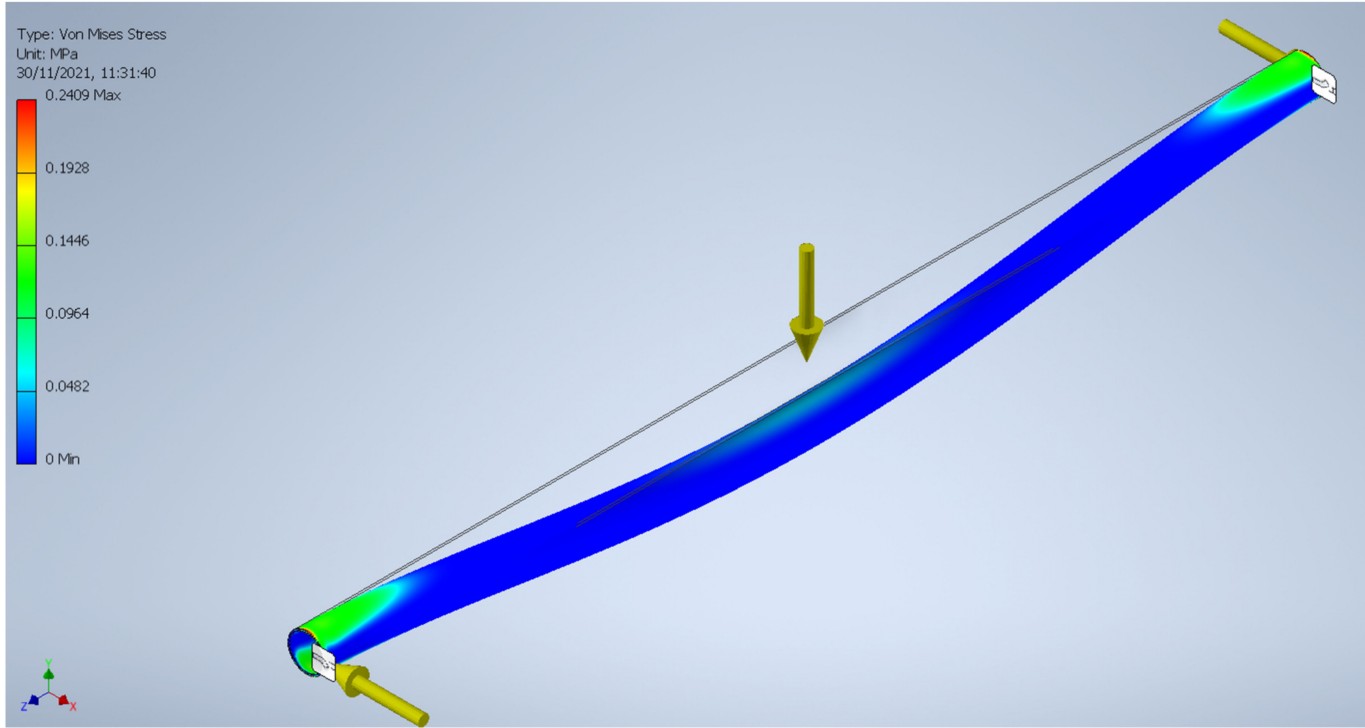

**Figure 11.** Bamboo simulation.

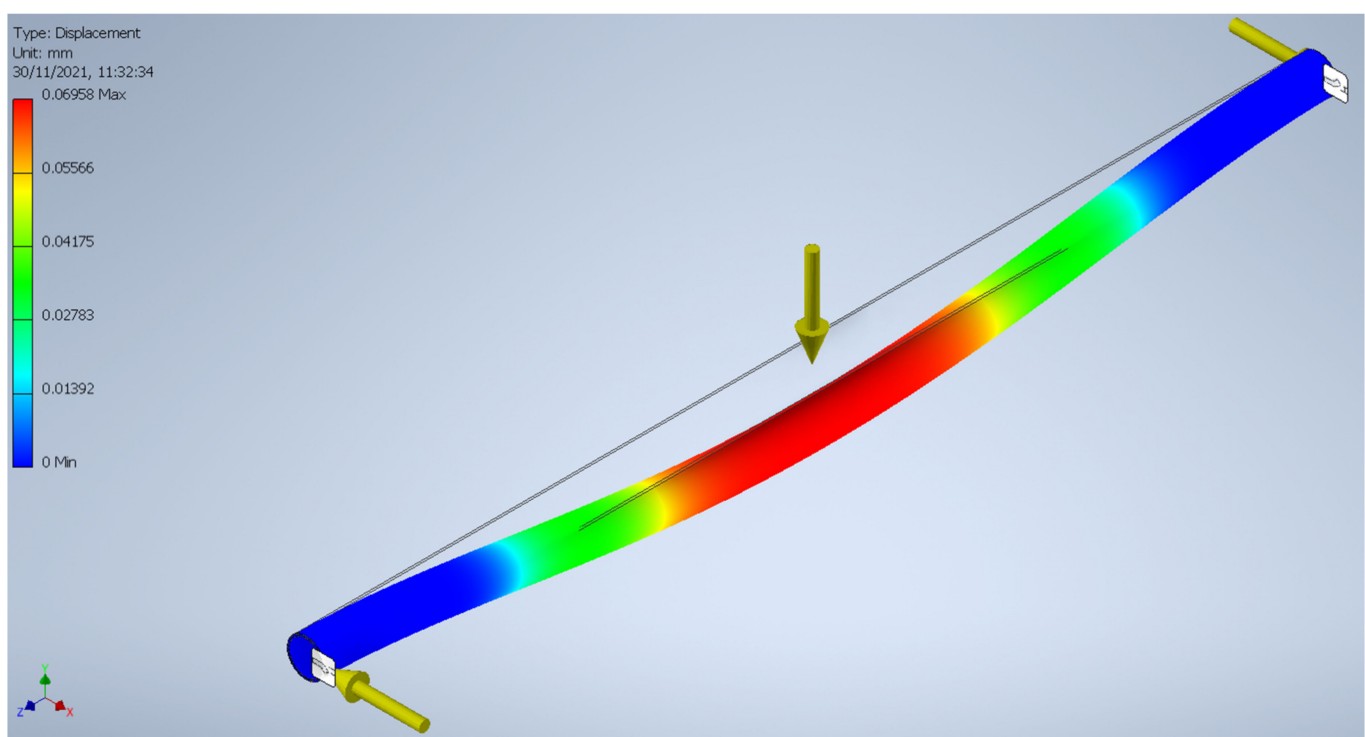

**Figure 12.** Guadua displacement.

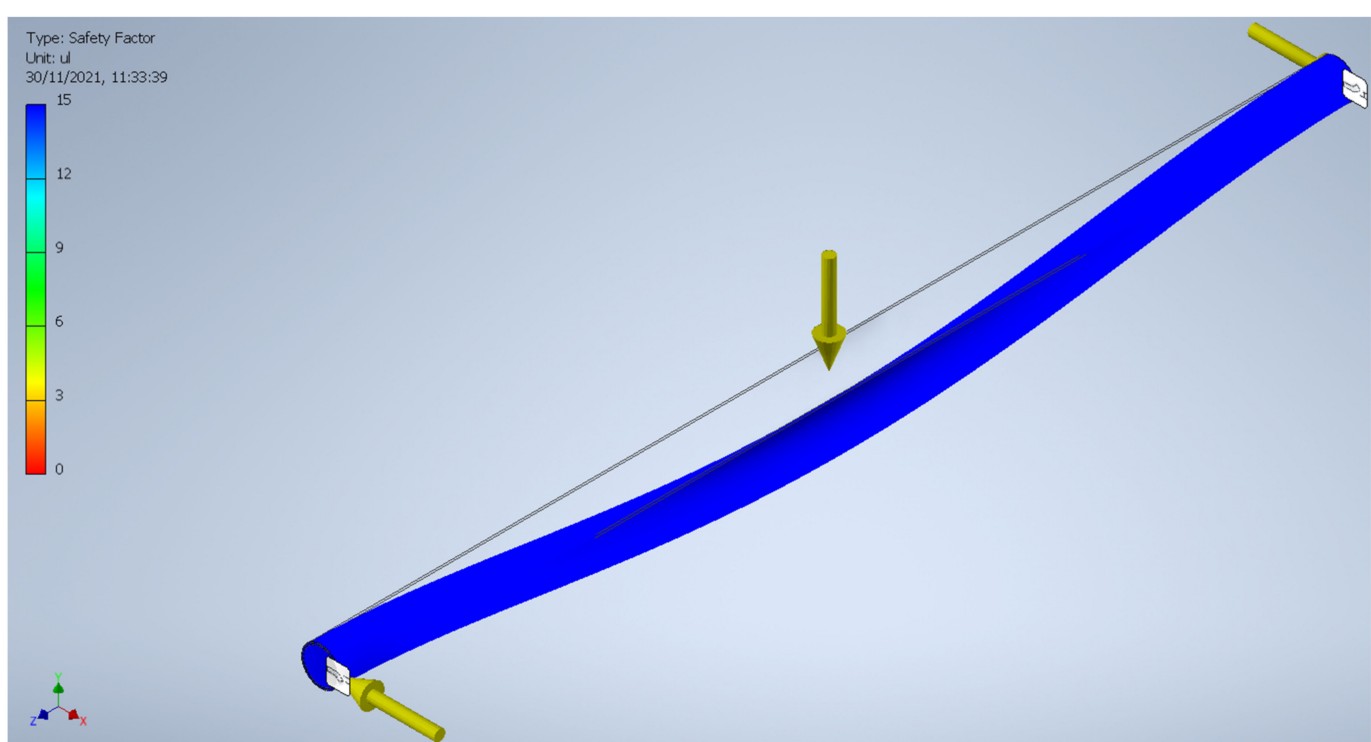

**Figure 13.** Guadua safety factor.

This result proves that normal use profiles in Latin America could be replaced by an alternative material such as bamboo and provide security for seismic events to the population; moreover, this material can be very useful in the construction of temporary shelter and social housing when needed.

## 4. Conclusions

In the search for a sustainable and alternative material to be used as a beam application on social interest housing, simulations were performed to analyze the behavior of the materials in Latin American conditions, where the utilization of specialized software showed that sustainable and accessible materials such as natural materials and reinforce concrete with natural fibers can be useful as beams to resist extreme forces in Latin America. However, since there are different useful options, the techniques of multicriteria decision method agreed that the best material for social interest housing is the bamboo also known as Guadua. Furthermore, with these results, it is possible to reaffirm that the Guadua is the best option for an alternative material for Latin American rural areas in construction, considering technical, environmental and price aspects, especially when it is most needed.

Since Latin America is placed under the Nazca plate, this has provoked some of the strongest earthquakes recorded, in this sense, every nation has developed different norms and codes for buildings to resist seismic events. In this way, for social interest housing a simulation was performed using the software CYPECAD, using the Ecuadorian normative, this simulation showed that the greatest load that a beam receives and must resist has a magnitude 12.17 KN, being this node of the building the weakest spot.

Furthermore, the need to develop buildings with sustainable and environmentally responsible materials has led to the search for alternative materials, nowadays informatic tools such as the CES-EDUPACK 2019 software allow us to have an extend library of materials that can be limited according to the developer's need. In this sense, the use of CES-EDUPACK 2019 threw several material options according to the specific need of resist earthquakes on Latin America; however, not all of them are suitable for the task, in this sense, fibers have important mechanical properties, but for itself cannot be used as beams; however, the same software with the synthesizer tool simulates the performance of the fibers as a reinforce for concrete, deploying a list of interesting sustainable and accessible materials.

Moreover, even though the results showed some alternatives to social interest housing in Latin America, there is the need to select the best of them. However, to make this selection the considerations of the material performance carry the technical, environmental and price aspects, in this sense, the MCDM allows us to assess different options and considers that the weight of each criterion is as important as the material selection itself, showing that the best material is the bamboo regarding all the before mentioned aspects. In this way, the bamboo has been shown to be a great material for construction around the world, but in this case, it was also shown to be the best compared with some other alternative materials by using mathematical and statistical techniques.

However, even though a selection methodology was implemented that included simulations, specialized software selection, bibliographical research and the utilization of operational tools, the best material must be validated for the proposed conditions. In this case, a finite element analysis was performed on a beam with the required characteristics, where it was found that the common species of bamboo in Latin America known Guadua Angustfolia, with a minimum diameter of 120 mm and a width of 5 mm, was shown to be enough to resist the loads that typical steel profile also must endure.

In this way, this research has selected the best material, nevertheless, it was limited to the utilization of the library on the CES EDUPACK 2019 software, which has a very important collection of materials information, but does not consider some native materials that are used on the region. In this sense, for future research, we recommend using a similar methodology as the one described here, but with a consideration of indigenous materials such as Pambil or banana fiber reinforced concrete to name any, this will allow us to expand the spectrum of knowledge on sustainable and environmentally responsible buildings.

**Author Contributions:** Conceptualization: J.M.-G. and J.F.N.; methodology: J.M.-G., J.F.N. and J.Y.; software: J.M.-G., J.F.N. and J.Y.; validation: J.M.-G., J.F.N. and J.Y.; formal analysis: J.F.N. and J.M.-G.; investigation: J.F.N. and J.M.-G.; resources: J.Y. writing—original draft preparation: J.F.N.;

writing—review and editing: J.M.-G.; visualization: J.F.N.; supervision: J.M.-G.; project administration: J.M.-G.; funding acquisition: J.M.-G. All authors have read and agreed to the published version of the manuscript.

**Funding:** This research takes part of the project Selection, characterization and simulation of phase change materials for thermal comfort, cooling and energy storage. This project is part of the INEDITA call for R&D research projects in the field of energy and materials. This research takes part of the project P121819, Parque de Energias Renovables funded by Universidad International SEK.

**Institutional Review Board Statement:** Not applicable.

**Informed Consent Statement:** Not applicable.

**Data Availability Statement:** The data reported in this research can be obtained from the corresponding author.

**Conflicts of Interest:** The authors declare that the research was conducted in the absence of any commercial or financial relationships that could be construed as a potential conflict of interest.

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
