# Peer review of "Selection of a Sustainable Structural Beam Material for Rural Housing in Latin América by Multicriteria Decision Methods Means"

_applsci, doi:10.3390/app12031393_

Round 1

Reviewer 1 Report

  1. The article states that the best beam material is selected (Chapter 2.4). What criteria are used to describe that it is the best material? The text of the article goes on to mention that criteria such as weight, modulus of elasticity, etc. are used. The choice of criteria is essential. Justify the choice of one or the other criteria.
  2. The impression is that the best material must be bamboo. This is related to the first remark.

Author Response

Dear reviewer,

Thank you for the time and work spent in the revision of our paper, it is very important for us to be considered for publication.

Regarding your observations. In chapter 2.4 we describe the method to select a preliminary list of materials that can be useful for the proposed application. This has been explained better in lines 241 to 243. However, the results of this selection are displayed in table 4 of chapter 3.2, where the order of the materials presented in table 4 is sorted alphabetically and does not reflect any preference. This issue has been explained better in lines 382 and 383.

Furthermore, after the generation of the candidate materials list with all the properties needed for the beam application, the materials are assessed by multicriteria decision methods. This issue has been explained better on lines 407 to 410. In this way, the multicriteria decision Methods of VIKOR, TOPSIS and COPRAS are used and all of them agrees that the best material is bamboo. This concept has been explained better on lines 484 to 488.

Once more let me extend our gratitude for your review and if there is aspect where we could improve, please let us know.

Best Regards.

Reviewer 2 Report

The authors have conducted a research on the Selection of a sustainable structural beam material for rural housing in Latin América using MCDM approaches. The research is interesting, while some improvements are needed before it can be published in Applied Science. My comments are as follows.

  • An overall flow of research should be presented at the beginning of the methodology section using a comprehensive research flowchart.
  • Section 2.6 should be revised. Presenting very short paragraphs under subsections is not appropriate and the explanations of each method are not adequate.
  • Most importantly, why did the authors adopt several MCDM techniques? And why these methods have been selected among all available MCDM methods? Clear justifications are required.
  • Also, how did you collect the data for each method? Who are the respondents? Please elaborate on every single method you have used.
  • The conclusion section should be revised. Please avoid writing very short paragraphs as each paragraph requires an introduction, body, and conclusion. Also, the limitation and future work should be added in this section.

Author Response

Dear reviewer,

Thank you for the time and work spent in the revision of our paper, it is very important for us to be considered for publication.

Regarding to your observations. At the beginning of the methodology some paragraphs have been added to make more comprehensive how the research was performed, this is described from lines to 109 to 144. Also, regarding the third point of your evaluation, the explanation of why these MCDM where chosen has been added to lines 120 to 142.

The sub section 2.6 has been corrected adding the governing equations of each method to make a better understanding of the multicriteria decision methods used for the calculations made.

The data for each method comes from the performance characteristics of each material, this numerical information obtained by the utilization of the software CES-EDUPACK are the data used in every method, in this sense there was no need to collect data outside the software. The explanation of this issues has been added on lines 255 to 259.

The conclusions have been rewritten according to your observations from lines 527 to 569. Also, the limitations and recommendations has been added from lines 570 to 577.

Once more let me extend our gratitude for your review and if there is aspect where we could improve, please let us know.

Best regards.

Round 2

Reviewer 2 Report

The authors have addressed my comments and the manuscript can be considered for publication